# The Characterization of the Neuroimmune Response in Primary Pterygia

**DOI:** 10.3390/ijms26157417

**Published:** 2025-08-01

**Authors:** Luis Fernando Barba-Gallardo, Sofía Guadalupe Ocón-Garcia, Manuel Enrique Avila-Blanco, José Luis Diaz-Rubio, Javier Ventura-Juárez, Elizabeth Casillas-Casillas, Martín Humberto Muñoz-Ortega

**Affiliations:** 1Health Science Center, Autonomus University of Aguascalientes, Aguascalientes C.P. 20100, Mexico; dra.sofia.ocon@gmail.com (S.G.O.-G.); elizabeth.casillas@edu.uaa.mx (E.C.-C.); 2Basic Science Center, Autonomus University of Aguascalientes, Aguascalientes C.P. 20100, Mexico; manuel.avila@edu.uaa.mx (M.E.A.-B.); javier.ventura@edu.uaa.mx (J.V.-J.); humberto.munozo@edu.uaa.mx (M.H.M.-O.); 3Board of Trustees of the Eye and Tissue Bank, Aguascalientes, Aguascalientes C.P. 20190, Mexico; joseluisdiazrubio@yahoo.com.mx

**Keywords:** neuroimmune, response, pterygium, inflammation, fibrogenic

## Abstract

Several chronic inflammatory processes are currently being studied in relation to other systems to better understand the regulation mechanisms and identify potential therapeutic targets. A significant body of evidence supports the role of the nervous system in regulating various immunological processes. This study investigates the relationship between pterygia and the sympathetic nervous system, focusing on their interaction in the inflammatory response and fibrogenic process. Sixteen surgical specimens of primary pterygia and four conjunctival tissue samples were examined, and their morphology was analyzed using hematoxylin–eosin and Masson’s trichrome stains. The gene expression of adrenergic receptors, as well as inflammatory and fibrogenic cytokines, was also assessed. Additionally, both adrenergic receptors and tyrosine hydroxylase were found to be localized within the tissues according to immunohistochemistry and immunofluorescence techniques. Increased expression of proinflammatory, fibrogenic, and adrenergic genes was observed in the pterygium compared to the healthy conjunctiva. Adrenergic receptors and tyrosine hydroxylase were localized in the basal region of the epithelium and within blood vessels, closely associated with immune cells. Neuroimmunomodulation plays a key role in the pathogenesis of pterygia by activating the sympathetic nervous system. At the intravascular level, norepinephrine promotes the migration of immune cells, thereby sustaining inflammation. Additionally, sympathetic nerve fibers located at the subepithelial level contribute to epithelial growth and the fibrosis associated with pterygia.

## 1. Introduction

A pterygium is an ocular surface condition characterized by the corneal invasion of the bulbar conjunctiva [1]. The global prevalence of pterygia is 10.2%, with higher rates observed in regions near the equator. Pterygia primarily affect adult males in rural populations, a pattern closely associated with outdoor activities [2]. The primary risk factor for pterygium development is ultraviolet (UV) radiation, which impacts limbal stem cells, creating a chronic inflammatory environment. This leads to the production of several key molecules, including interleukin 6 (IL-6), interleukin 8 (IL-8), basic fibroblast growth factor (bFGF), fibroblast growth factor beta (FGF-β), platelet-derived growth factor (PDGF-B), transforming growth factor beta 2 (TGF-β2), and tumor necrosis factor alpha (TNF-α). In addition, the downregulation of interferon beta (IFN-β) contributes to the recruitment of leukocytes, sustaining inflammation and promoting connective tissue remodeling. This process enhances conjunctival scarring by increasing fibroblasts (originating from perivascular connective tissue) and myofibroblasts (from resident fibroblasts). Furthermore, UV-induced epithelial damage stimulates the formation of new blood vessels, which is mediated by vascular endothelial growth factor (VEGF), metalloproteinase-9 (MMP-9) and nitric oxide (NO) [2,3,4].

Immunological processes are intricately regulated by various mechanisms, one of which is neuroimmunomodulation, a process involving the interaction between the immune and nervous systems. This reciprocal communication is facilitated by neurotransmitter receptors such as those for noradrenaline and acetylcholine on immune cells, as well as the innervation of lymph nodes by sympathetic nervous system (SNS) fibers. Neuroimmunomodulation plays a key role in regulating immune responses. Given that chronic inflammation underlies the pathogenesis of fibrotic processes, the neuroimmunomodulatory functions of both the parasympathetic and sympathetic nervous systems may hold significant clinical implications [5].

The sympathetic nervous system, through norepinephrine, plays a key role in regulating and progressing various fibrotic diseases, including pulmonary fibrosis, heart failure, systemic sclerosis, chronic kidney disease, and hepatocellular carcinoma. Evidence suggests that norepinephrine and epinephrine, via the beta-2 adrenoreceptor, can inhibit the production of type-1 proinflammatory cytokines, such as IL-12, TNF-α, and IFN-γ, via antigen-presenting cells and Th1 helpers. In contrast, this pathway stimulates the production of type-2 anti-inflammatory cytokines, such as IL-10 and TGF-β. Through this mechanism, endogenous catecholamines can selectively suppress Th1 responses and cellular immunity, promoting a Th2 shift that favors humoral immunity. On the other hand, norepinephrine, acting through alpha-1 adrenergic receptors, enhances the function of immune cells, particularly NK cells, macrophages, and T and B lymphocytes, thereby amplifying both the inflammatory and fibrogenic processes [6,7,8,9,10,11].

However, its role in the pathogenesis of pterygia remains unclear. Therefore, this study aims to investigate the relationship between pterygia and the sympathetic nervous system, focusing on the neuroimmune interactions involved in the inflammatory and fibrogenic processes associated with this condition.

## 2. Results

### 2.1. Characteristics of the Study Subjects

Sixteen patients with primary pterygium were studied, consisting of 68.8% men and 31.3% women, aged between 39 and 67 years. The comorbidities observed included 31.3% with hypertension, 18.8% with diabetes, 12.5% with both hypertension and diabetes, 6.3% with hypothyroidism, and 31.3% who were healthy. The control group, consisting of four individuals (three men and one woman), aged between 46 and 52 years, had no comorbidities.

### 2.2. Histopathology Description of the Tissue

The macroscopic characteristics of a pterygium are shown below (Figure 1). It appears as a triangular-shaped tissue with regular edges, and the average diameter of the samples was 7.27 ± 2.15 mm. The tissue is pale pink and has a rubbery consistency once extracted from the ocular surface, often tending to roll along its axis. Its interior displays branching blood vessels of varying calibers.

Histological sections compare healthy conjunctiva and a pterygium. In Figure 1B, panels α, β, and γ, conjunctiva sections are shown, with the following characteristics: non-keratinized, stratified squamous epithelium with melanocytes at its base and a thickness of approximately 10 cells. Beneath the epithelium is the stroma, composed of loose connective tissue, fibroblasts, and small-caliber blood vessels. A morphological analysis of all samples revealed that, based on their predominant component, 45.5% were angiomatous, 36.4% were fibrous, and 18.2% were mixed. The angiomatous type (δ) exhibited a disorganized, flat, non-keratinized stratified epithelium with significant hypertrophy of goblet cells. The stromal layer showed multiple large blood vessels. The fibrous type (ε) was characterized by a highly disorganized, hyperplastic, non-keratinized stratified epithelium up to 20 cells thick, with mild goblet cell hypertrophy and a thickened, fibrotic stromal layer. Additionally, squamous metaplasia and keratinization were observed. A significant inflammatory infiltrate was noted in the mixed type, which combined features of both the angiomatous and fibrous types (ζ), particularly in the perivascular region of the tissue. Masson’s trichrome staining (Figure 2) revealed that the pterygium predominantly consists of collagen fibers in the subepithelial and stromal regions.

### 2.3. Localization of Tyrosine Hydroxylase and Adrenergic Receptors in the Pterygium

Figure 3 illustrates the localization of tyrosine hydroxylase (TH) and adrenergic receptors in the pterygium using immunohistochemistry. In Panel A, TH expression is shown through immunoperoxidase staining in a composite image of the entire tissue (X10), revealing its presence predominantly in the subepithelial zone as filamentous structures. Higher magnifications (X40 and X100) in Figures α and β highlight this subepithelial localization (blue indicator), while Figure γ (X40) demonstrates TH staining around blood vessels. In Panel B, the beta-2 adrenergic receptor (ADRB2) is visualized using a similar methodology. The composite image (X10) confirms a widespread distribution, with Figure α (X100) showing ADRB2 expression in the subepithelial region (blue indicator) and perivascular areas (arrow). Additionally, Figure β reveals the presence of alpha-1 adrenergic receptor (ADRA1) within the vascular smooth muscle layer (X100).

### 2.4. Immune Response in the Pterygium and Conjunctiva

Using immunofluorescence, we analyzed the presence of immune response cells (Figure 4B,D) in both the pterygium and conjunctiva. We observed a significant increase in the pterygium, with CD3-positive T lymphocytes showing a 1148 ± 175.4-fold increase (*p* < 0.001) and CD68-positive macrophages showing a 587.0 ± 53.99-fold increase (*p* < 0.0001) compared to the healthy conjunctiva (Figure 4A,C).

### 2.5. Neuroimmune Relationship in the Pterygium

Using double immunofluorescence, we examined the interaction between immune cells and the sympathetic nervous system, uncovering a strong interrelationship between TH, lymphocytes (Figure 5A), and macrophages (Figure 5B). This finding suggests a neuroimmune interaction in the pterygium that is similar to those described by Oben and Neuhuber [12,13] in models of hepatic fibrosis.

### 2.6. Inflammatory, Fibrogenic, and Adrenergic Markers’ Gene Expression

Gene expression analysis (Figure 6, Panel A) revealed significant overexpression of inflammatory genes in the pterygium, with differences up to 24.7-fold for IL17A (*p* < 0.05) and 2.8-fold for TNF-α (*p* < 0.05) compared to healthy conjunctiva. Regarding fibrosis-related genes, a 5.8-fold overexpression (*p* < 0.05) of the COL4A1 gene was observed compared to the conjunctiva. No significant differences were found in the expression of TGFB1 (*p* > 0.05).

Figure 6 Panel (B) exhibited a marked increase in expression. Specifically, the expression of TH increased 14.3-fold (*p* < 0.05), while the adrenergic receptors ADRA1A, ADRA1B, and ADRB2 showed increases of 3.4-fold (*p* < 0.05), 1.5-fold (*p* < 0.01), and 1.4-fold (*p* < 0.01), respectively, compared to the healthy conjunctiva (Figure 6).

## 3. Discussion

The tissue characteristics observed in this study are consistent with those described by several authors, aligning with the histological descriptions [14,15]. However, this study also analyzed the clinical appearance through ophthalmological assessment using slit-lamp examination to identify angiogenic and fibrotic differences observed in the histological sections.

There is a considerable bibliographic background of extensive analyses of the cellular and genetic biomarkers in pterygia. However, a thorough search revealed that there is still a lack of studies addressing the type of innervation in the pterygium and its clinical manifestations, especially concerning its fibrotic or angiomatous features. Therefore, this research group employed molecular and cellular techniques to demonstrate the presence and expression of receptors associated with the sympathetic system and their possible relationship with inflammatory cells and areas of fibrosis. As a result, this study provides evidence, through immunofluorescence and molecular analysis, that allows us to generate knowledge and suggest that there is a relationship between the sympathetic nervous system and the histopathological process observed in pterygia, similar to the relationships described by Oben (2004) [14] and Neuhuber (2004) [15] in models of hepatic fibrosis, where there is a relationship between fibers of the autonomic nervous system with immune response cells such as Kupfer and fibrogenic cells such as hepatic stellate cells.

Our results confirm the presence of ADRA1A and ADRB2 receptors in the conjunctiva, suggesting that the predominant innervation is sympathetic. Heukels et al. [16] mentioned that corneal innervation might share similar functions to those of the ophthalmic branch of the trigeminal nerve (V1) and certain maxillary nerves (V2). Accordingly, the cornea plays roles related to trophic influences on the corneal epithelium, such as surface desiccation due to reduced tear secretion; decreased corneal sensitivity, leading to a weakening of the protective blink reflex; abnormal epithelial cell metabolism, resulting in an inability to resist trauma, desiccation, and infections; and the loss of trophic support provided by corneal nerve fibers [17].

While the cornea is primarily innervated by adrenergic fibers, the limboscleral region is mainly dominated by cholinergic endings [18]. This suggests that the adrenergic receptors found in the pterygium likely originate from the corneal nerves rather than from the cholinergic limbal endings. In addition, the presence of Krause’s terminal corpuscles, with their simple anatomic structure, suggests that they can easily extend into the pterygium. However, due to the nature of their pathway, the entry or continuity of nerve endings directed toward the pterygium is also unlikely, so further experiments are needed to elucidate the possible origin of these nerve fibers innervating the pterygium [19].

The increase in the expression of IL17, TNF-α, and COL4A1 demonstrates the involvement of immune cells whose inflammatory and fibrotic cytokines—TNF-α produced by macrophages and monocytes and IL17A produced by lymphocytes—regulate the deposition of extracellular matrix in tissues [3,20,21]. TGFB1, a profibrotic cytokine [22,23,24,25,26,27], did not show significant overexpression on average across cases. However, some individual cases exhibited increased expression, likely due to the variability in the histological types of pterygium studied [4].

The increased expression of TH, ADRA1A, ADRA1B, and ADRB2 suggests the involvement of the sympathetic system in the pathogenesis of pterygium. These findings align with those of several authors who have demonstrated the role of epinephrine in fibrogenesis across several organs, including the lung, heart, skin, kidney, and liver [8,9,10,11].

Based on this evidence, Tiegs et al. (2004) has reported that the distribution of nerve fibers and immune cells in the liver is not random, and partial overlap of these patterns may promote neuroimmune interactions, as a relationship of tyrosine hydroxylase-positive sympathetic nervous system fibers with immune response cells and hepatic stellate cells has been found [15].

Norepinephrine and epinephrine are activators of immune cell functions, mainly NK cells, macrophages and T and B lymphocytes. In humans, injection of catecholamines causes an increase in NK cell migration in the first 2 to 4 h although, over time, it causes a decrease in their functionality and causes granulocytes to be attracted to specific sites. Acquired cellular immunity is affected by catecholamines in a way that inhibits the Th1 response and stimulates the Th2 response (humoral immunity) when there is a beta-adrenergic effect while triggering the Th1 and innate inflammatory immune response when there is an alpha-adrenergic effect [28].

Similarly to the distribution observed by Tiegs et al. [13] in the liver, we identified a potentially neuroimmune interaction pattern in pterygium samples. This interaction occurs between tyrosine hydroxylase–positive fibers, associated with the sympathetic nervous system, and immune response cells marked by CD3 and CD68. It is known that these immune cells express catecholamine receptors, as observed by Elenkov et al. [28]. Furthermore, during inflammatory processes, sympathetic hyperactivity leads to increased catecholamine release, which could promote the proliferation of immune cells found in pterygium samples, as well as the overexpression of the evaluated cytokines.

### Limitations

Limitations statement low number of conjunctiva analysis for genetic testing (only 4 conjunctiva) and lack of proper control

## 4. Material and Methods

### 4.1. Patients

Surgical specimens of primary pterygium (n = 28) were obtained from patients undergoing pterygium resection surgery, while surgical specimens of normal conjunctiva (n = 4) were collected from patients undergoing glaucoma surgery and patients undergoing enucleation surgery, and without any treatment with tears. These tissues were generously donated to the Autonomous University of Aguascalientes (UAA) for research. Patients did not receive ocular treatments with steroids or topical vasoconstrictors within one week prior to surgery. Additionally, patients with concurrent inflammatory ocular surface pathology were excluded from the study. Informed consent was obtained from all participants prior to their inclusion. All procedures were conducted in accordance with the principles outlined in the Declaration of Helsinki and the institutional guidelines of the UAA (UAA-BIOETICA-OCT2023).

### 4.2. Sample Selection

Of the 28 specimens that met the inclusion criteria, 4 pterygium surgical specimens were excluded due to insufficient tissue. The remaining specimens were randomly assigned into two groups: one for gene expression analysis (comprising 8 pterygium and 8 conjunctiva tissues), and the other for a histopathological study (comprising 8 pterygium and 8 conjunctiva tissues). The tissues designated for gene expression analysis were stored in RNAlater^®^ at −81 °C (Invitrogen, Carlsbad, CA, USA), while those allocated for histopathological analysis were preserved in 2% paraformaldehyde until further study. Due to the nature of the tissue, it was not possible to use the same sample for both analyses.

### 4.3. Histopathological Analysis

#### 4.3.1. Histological Sections

For histopathological analysis, the samples initially stored in 2% paraformaldehyde were washed three times with 1X phosphate-buffered saline (PBS). The tissues were then fixed using Tissue-Tek^®^ (Sakura Finetek, Torrance, CA, USA) and liquid nitrogen. Subsequently, 5-micron sections were prepared at −25 °C using a cryostat (Leica Biosystems, Chicago, IL, USA), with the samples placed on silanized slides. The tissue specimens were stored at 4 °C for preservation.

#### 4.3.2. Morphological Analysis

To analyze the histological characteristics of the samples, hematoxylin–eosin staining was combined with Masson’s trichrome staining to assess collagen fibers [5]. The histological preparations were examined using an Axioskop 40/40 FL polarized light microscope (Carl Zeiss AG) and analyzed with Image-Pro Plus Software 4.5.1 (Media Cybernetics, Inc. Rockville, MD, USA). The percentage of fibrosis was calculated as the ratio of the fibrotic area to the total tissue area.

#### 4.3.3. Immunohistochemistry

The following antibodies and dilutions were used for the immunohistochemistry tests: Beta-2 Adrenergic Receptor antibody at a dilution of 1:100 (MAB10040-100, Novus Biologicals, Denver, CO, USA) and a secondary antibody, Goat Anti-Rabbit IgG Antibody HRP conjugate, at a dilution of 1:200 (AP307P, Merck, Rahway, NJ, USA). Detection was achieved using 3,3′-diaminobenzidine (DAB) at a dilution of 1:10 (Thermo Fisher, Waltham, MA, USA), and the procedure was repeated, this time incubating all slides overnight with the primary antibody Anti-Tyrosine Hydroxylase at a dilution of 1:200 (SAB4502966, Sigma-Aldrich, St. Louis, MO, USA). Following a second wash with PBS-Triton X-100 (0.2%), the slides were incubated with secondary antibodies, goat Anti-Rabbit IgG Antibody HRP conjugate, at a dilution of 1:200 (AP307P, Merck, Rahway, NJ, USA).

#### 4.3.4. Immunofluorescence

Indirect immunofluorescence was conducted to identify the presence of macrophages (CD68^-^positive cells) and T lymphocytes (CD3^-^positive cells) in the tissues. The primary antibodies used were Monoclonal Mouse Anti-Human CD3 at a dilution of 1:250 (M0756, Dako, Santa Clara, CA, USA) and Monoclonal Mouse Anti-Human CD68 at a dilution of 1:250 (M0814, Dako, Santa Clara, CA, USA). The secondary antibody used was Goat Anti-Mouse IgG Alexa Fluor 594^®^ at a dilution of 1:1000 (A11005, Invitrogen, Carlsbad, CA, USA). Nuclei were counterstained with Hoechst 33258^®^ (Invitrogen, Carlsbad, CA, USA) [29].

#### 4.3.5. Immunofluorescence Double Labeling

To assess the colocalization of the TH enzyme, macrophages, and T lymphocytes, double-labeled indirect immunofluorescence was performed following the method of [14]. One set of slides was incubated overnight at room temperature in a humid chamber with the primary antibody Monoclonal Mouse Anti-Human CD3 at a dilution of 1:250 (M0756, Dako, Santa Clara, CA, USA), while the other set was incubated with the primary antibody Monoclonal Mouse Anti-Human CD68 at a dilution of 1:250 (M0814, Dako, Santa Clara, CA, USA). After incubation, the slides were washed with 0.2% PBS-Triton X-100, and the procedure was repeated, this time incubating all slides overnight with the primary antibody Anti-Tyrosine Hydroxylase at a dilution of 1:200 (SAB4502966, Sigma-Aldrich, St. Louis, MO, USA). Following a second wash with PBS-Triton X-100 (0.2%), the slides were incubated with secondary antibodies, including Goat Anti-Mouse IgG (H+L) Cross-Adsorbed Secondary Antibody, Alexa Fluor 594^®^ at a dilution of 1:1000 (A11005, Invitrogen, Carlsbad, CA, USA) for red-orange labeling of CD3 and CD68, and Goat Anti-Mouse IgG (H+L) Highly Cross-Adsorbed Secondary Antibody, Alexa Fluor Plus 488^®^, at a dilution of 1:1000 (A32723, Invitrogen, Carlsbad, CA, USA) for green labeling of TH. Nuclei were counterstained with Hoechst 33258^®^ (Invitrogen, Carlsbad, CA, USA).

#### 4.3.6. Image Analysis

Tissues were examined using an Axioskop 40 and an Axiovert 40 CFL microscope (Zeiss, Oberkochen, Germany). Images were analyzed using AxioVision software (Zeiss, Oberkochen, Germany). A field sweep of the entire sample was conducted for the composite images. All images were processed using Fiji software Ver. 2.16.0/1.5p (Fiji Is Just ImageJ, GNU General Public License).

### 4.4. Gene Expression Analysis

The relative expression of genes related to inflammation (interleukin 17A (IL17A)) and tumor necrosis factor alpha (TNF-α)), fibrosis (transforming growth factor beta 1 (TGFB1) and collagen 4A1 (COL4A1)), and adrenergic response (tyrosine hydroxylase (TH), the enzyme precursor of epinephrine and norepinephrine, and the adrenergic receptors alpha 1A [ADRA1A], alpha 1B [ADRA1B], and beta 2 [ADRB2])) was evaluated using real-time quantitative polymerase chain reaction (qPCR). The primers used are listed in Table 1. Messenger ribonucleic acid (mRNA) was extracted according to the protocol outlined in the Quick-RNA MiniPrep Plus Kit (Zymo Research, Tustin, CA, USA), maintaining the cold chain throughout. mRNA quantification was performed using Biodrop (Isogen Life Science, Barcelona, Spain), adjusting the mRNA volume of all samples to 1 μg. Reverse transcription was carried out following the GoScript^®^ protocol (Promega A5000, Madison, WI, USA). Subsequently, qPCR was performed using the SYBR Green/ROX qPCR Master Mix (2x) Max Kit (K022.1, Thermo Scientific, Waltham, MA, USA) on a StepOneTM system (Applied Biosystems, Waltham, MA, USA) under the following thermocycling conditions: initial denaturation at 95 °C for 3 min, denaturation (40 cycles) at 95 °C for 40 s, annealing (40 cycles) at 60 °C for 45 s, denaturation (MELT curve) at 95 °C for 15 s, annealing (MELT curve) at 60 °C for 1 min, and final denaturation (MELT curve) at 95 °C for 15 s. The relative expression levels were normalized to the constitutive gene glyceraldehyde-3-phosphate dehydrogenase (GAPDH), and differences were determined using the ΔΔCt method [30,31].

### 4.5. Statistical Analysis

Statistical analysis was conducted using Prism 8.0.1 software (GraphPad, Boston, MA, USA). The normality of the data was first assessed with the Shapiro–Wilk test. Groups (pterygium and conjunctiva) were then compared using Student’s t-test for normally distributed data and the Mann–Whitney U test for non-parametric data. Differences were considered statistically significant when *p* < 0.05.

### 4.6. Use of Artificial Intelligence

In this study, artificial intelligence, specifically ChatGPT 4.0, was used to review and improve English grammar.

## 5. Conclusions

Neuroimmunomodulation by the sympathetic nervous system could influence the activity in the pathogenesis of pterygia. This process occurs both at the intravascular level, where it promotes the migration of macrophages and lymphocytes, and within the nerve fibers in the subepithelial region of the tissue, where it contributes to epithelial hyperplasia and fibrosis. These findings pave the way for future studies focusing on the sympathetic innervation of the pterygium. Finally, the topical application of adrenergic receptor antagonists at the ocular level could potentially serve as a therapeutic strategy to slow the progression and growth of pterygia.

## Figures and Tables

**Figure 1 ijms-26-07417-f001:**
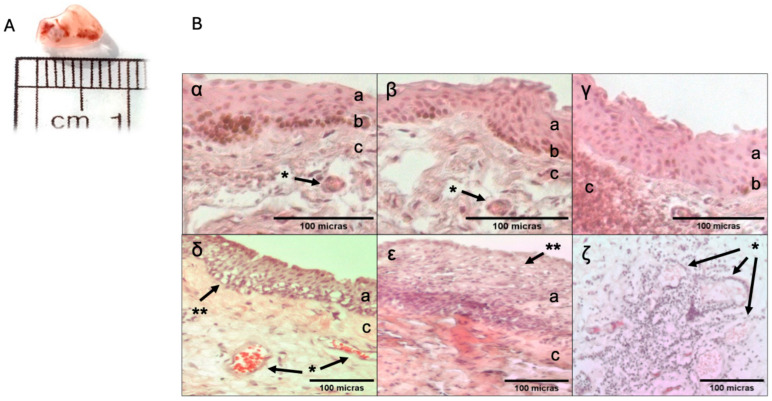
Panel (**A**): Macroscopic photograph of the pterygium. The photograph of the tissue was taken with a digital camera, using a ruler to show its length (1X). Panel (**B**): comparative analysis of healthy conjunctiva and pterygium. The conjunctiva (α–γ) (HEALTHY) has an epithelial layer (a), melanocytes in the basal epithelial zone (b), and a stromal layer (c) with some blood vessels (*). The pterygium (δ–ζ), with its angiomatous (γ) and fibrous (ε) histological types, shows morphological changes in its epithelial (a) and stromal (c) layers, with proliferation of blood vessels (*) and hypertrophy of goblet cells (**). Image (ζ) shows the presence of inflammatory infiltrate in the pterygium in the region near the blood vessels (*). Hematoxylin–eosin staining. Figures (α–γ) (X20)—Axiovert 40 cfl microscope (Zeiss). Figures (δ–ζ) (X10)—Axioskop 40 microscope (Zeiss). Image analysis was performed using AxioVision (Zeiss) and FIJI (Fiji Is Just Image J, GNU General Public License).

**Figure 2 ijms-26-07417-f002:**
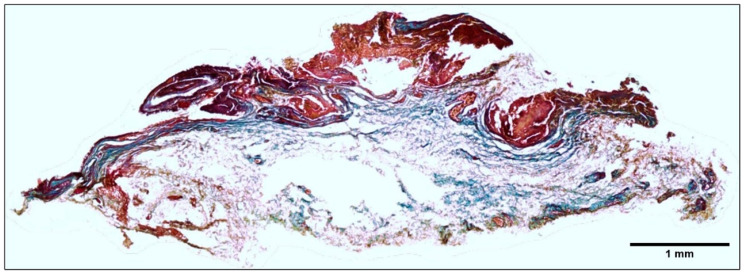
The analysis of the pterygium’s composition. The pterygium was composed of 90% collagen fibers (blue) located below the epithelial layer (red). Masson’s trichrome stain (composite image) was used field by field in the whole tissue (X10), with an Axioskop 40 microscope (Zeiss). Image analysis was performed using AxioVision (Zeiss) and FIJI (Fiji Is Just Image J, GNU General Public License).

**Figure 3 ijms-26-07417-f003:**
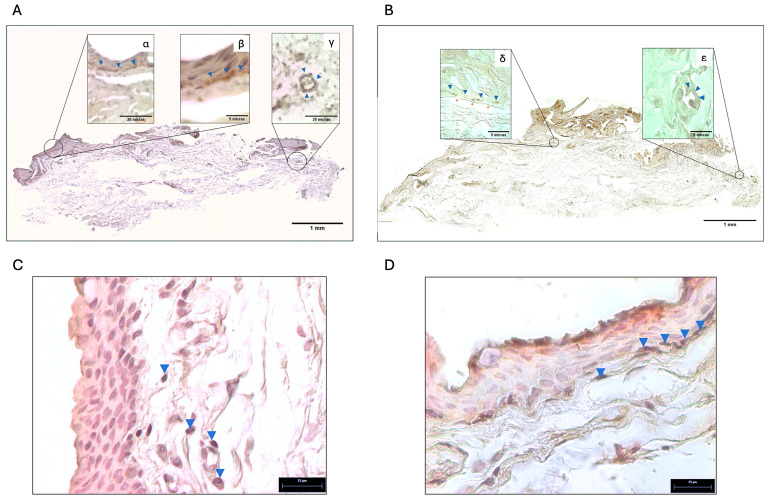
Panel (**A**): Localization of tyrosine hydroxylase (TH) in pterygium tissue. Immunohistochemistry was performed using the immunoperoxidase method with diaminobenzidine (DAB) as chromogen, and images are presented as a composite from multiple fields at 10× magnification. Images (α) and (β) show higher magnifications at 40× and 100×, respectively, highlighting subepithelial localization of TH (blue arrowheads). Image (γ) shows vascular localization of TH at 40× magnification (blue arrowheads). Panel (**B**): Localization of the β2-adrenergic receptor (ADRB2) in pterygium. Immunohistochemistry with DAB was performed and presented as a composite image from multiple fields at 10× magnification. Image (δ) shows high magnification (100×), revealing both subepithelial (blue arrowheads) and perivascular (red arrows) localization of ADRB2. Image (ε) shows localization of the α1-adrenergic receptor (ADRA1) in perivascular smooth muscle at 100× magnification (blue arrowheads). Panel (**C**): Localization of TH in control conjunctival tissue. Immunohistochemistry with DAB was performed. Blue arrowheads indicate TH-positive cells in perivascular areas, suggesting the presence of sympathetic nerve fibers (40× magnification). Panel (**D**): Localization of ADRB2 in control conjunctival tissue. Immunohistochemistry with DAB was performed. Blue arrowheads indicate TH-positive cells in subepithelial areas, consistent with sympathetic adrenergic activity in the epithelium (40× magnification). All images were acquired using an Axioskop 40 microscope (Zeiss), and image processing and analysis were conducted using Axio-Vision (Zeiss) and FIJI (Fiji Is Just ImageJ, GNU General Public License).

**Figure 4 ijms-26-07417-f004:**
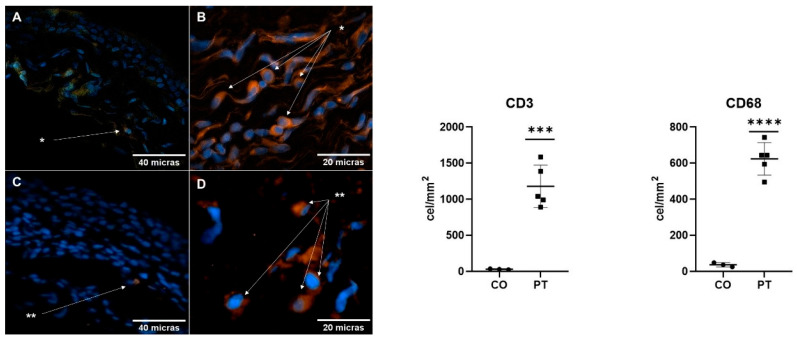
The immune response in the conjunctiva and pterygium. A comparative image shows the immune response in healthy pterygium and conjunctiva. Immunofluorescence: labeling with Alexa Fluor 594^®^ (red-orange), nuclei labeling with Hoechst 33258^®^ (blue). Figures (**A**) (conjunctiva X20) and (**B**) (pterygium X40) show positivity for T lymphocytes (CD3), marked as *. Panel (**C**) (conjunctiva X20) and (**D**) (pterygium X40) show positivity for macrophages (CD68), marked as **. Axioskop 40 microscope (Zeiss). Image analysis was performed with AxioVision (Zeiss) and FIJI (Fiji Is Just Image J, GNU General Public License). The immune cell count in the conjunctiva and pterygium is shown. The cell count of CD3-positive T lymphocytes and CD68-positive macrophages in histological sections of healthy conjunctiva (CO) and pterygium (PT) is shown. The dots determine the cellular quantification of each patient. The mean ± standard error of the mean (SEM) is also shown; ***: *p* < 0.001, ****: *p* < 0.0001.

**Figure 5 ijms-26-07417-f005:**
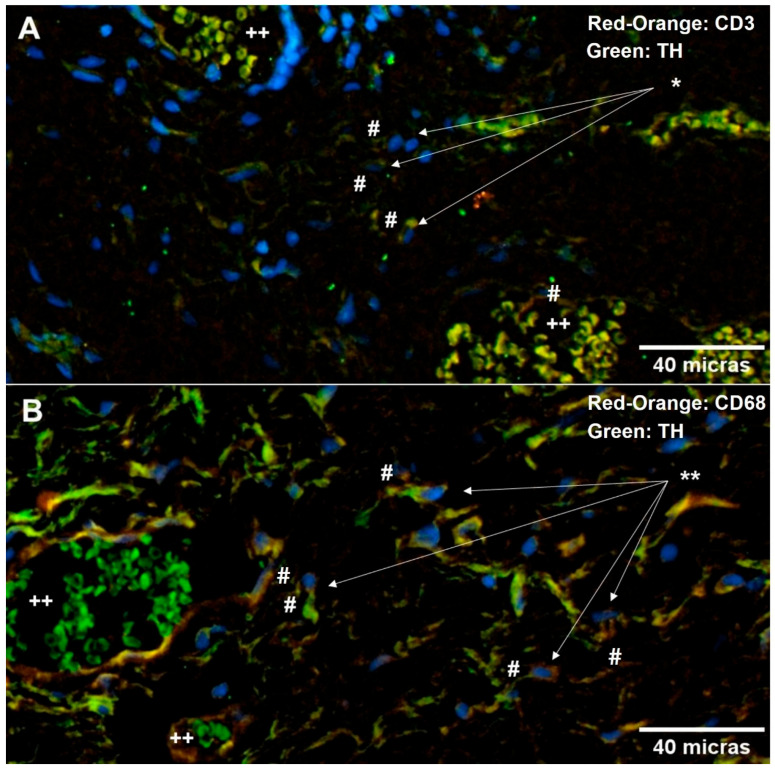
The neuroimmune relationship in the pterygium. The image shows the immune cell infiltrate present in the pterygium and its close relationship with TH (X20). Double immunofluorescence: labeling of macrophages and lymphocytes was performed using Alexa Fluor 594^®^ (red-orange), labeling of TH was performed using Alexa Fluor 488^®^ (green), and labeling of nuclei was performed using Hoechst 33258^®^ (blue). Panel (**A**) shows CD3-positive T lymphocytes (*), and panel (**B**) shows CD68-positive macrophages (**) interacting with HT (#). Some erythrocytes were observed in the blood vessels (++). Images were taken using an Axioskop 40 microscope (Zeiss). Image analysis was performed using AxioVision (Zeiss) and FIJI (Fiji Is Just Image J, GNU General Public License).

**Figure 6 ijms-26-07417-f006:**
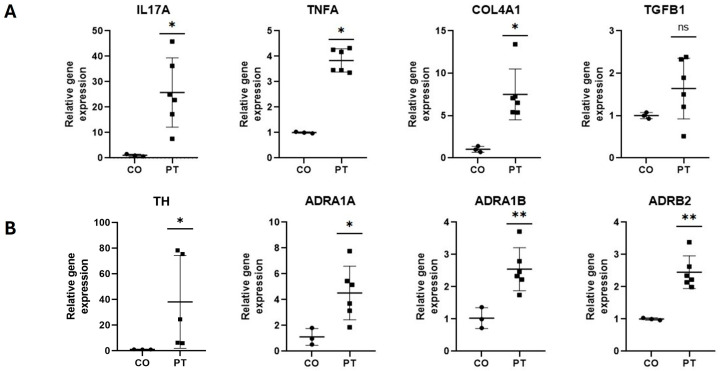
Panel (**A**): The relative expression of inflammatory (IL17A and TNF-α) and fibrogenic (COL4A1 and TGFB1) genes in healthy conjunctiva (CO) and pterygium (PT). The dots indicate the gene expression of each patient. Panel (**B**): The relative expression of adrenergic genes (TH, ADRA1A, ADRA1B, and ADRB2) in healthy conjunctiva (CO) and pterygium (PT). The dots indicate the gene expression of each patient. Mean ± standard error of the mean (SEM) are also shown; *: *p* < 0.05, **: *p* < 0.01.

**Table 1 ijms-26-07417-t001:** The gene sequence of the oligonucleotides.

Primer Name	Sequence (5′-3′)	Target Gene
Forward	5′-ACC AAT CCC AAA AGG TCC TC-3′	IL-17
Reverse	5′-GGG GAC AGA GTT CAT GTG GT-3′	
Forward	5′-CTC TTC TGC CTG CTG CAC TTT G-3′	TNF-α
Reverse	5′-ATG GGC TAC AGG CTT GTC ACT C-3′	
Forward	5′-CCC AGC ATC TGC AAA GCT C-3′	TGFB
Reverse	5′-GTC AAT GTA CAG CTG CCG CA-3′	
Forward	5′-CTG GTC CAA GAG GAT TTC CA-3′	COL4A1
Reverse	5′-TTT TGG TCC CAG AAG GAC AC-3′	
Forward	5′-GTG TTC CAG TGC ACC CAG TA-3′	TH
Reverse	5′-AGC GTG GAC AGC TTC TCA AT-3′	
Forward	5′-TGG CCG ACC TCC TGC TCA CCT C-3′	ADRA1A
Reverse	5′-GGC CCC GGC TCT CCC TCT TG-3′	
Forward	5′-CCC CCG ACG CCG TGT TCA AGG TG-3′	ADRA1B
Reverse	5′-CTG AGG CGC GGG CAG GCT CAG GA-3′	
Forward	5′-GAT TTC AGG ATT GCC TTC CA-3′	ADRB2
Reverse	5′-AAA TAA TGT ACG GGG GAG ATG CAT GA-3′	
Forward	5′-GAG TCA ACG GAT TTG GTC GT-3′	GAPDH
Reverse	5′-GAC AAG CTT CCC GTT CTC AG-3′	

## Data Availability

Dataset available on request from the authors.

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
