# Peer review of "The Characterization of the Neuroimmune Response in Primary Pterygia"

_ijms, 2025, doi:10.3390/ijms26157417_

Round 1
Reviewer 1 Report
Comments and Suggestions for Authors
See the Attached File

Reviewer 2 Report
Comments and Suggestions for Authors
The authors present a potential interesting research on pterygium. However, there are a few draw-backs:
The 4 conjunctiva (controls) samples were taken from glaucoma patients. However, the indication for surgery in glaucoma are cases that do not respond well to drops. The inflammation generated by the antiglaucomatous drops could change the conjunctiva. Could the authors exclude this possibility?
Figure 5 - What does C represent on the photos? `Red-orange: C`
Line 355 Please correct punctuation
Line 254. Figure 4 B and D represent only pterygium (not conjunctiva)? Please correct
The analysis of interactions between simpathethic fibers and immune cells is not clear. IT would be necessary for the authors to present integrative data from all 16 pterygium specimens and to compare it to normal conjunctiva. 4 normal controls are not enough to provide statistical relevance.
Author Response
Reviewer 2: Please see the attachement

Round 2
Reviewer 1 Report
Comments and Suggestions for Authors
Accepted
Author Response
"Please see the attachment

Reviewer 2 Report
Comments and Suggestions for Authors
The autors present only a slight revised form of the study. However, the choice of controls is still not adequate. Angle closure glaucoma presents with very high inflammation, congestion etc and is not compatible to normal conjunctiva.
I suggest the authors to have conjunctival tissue from enucleated eye (eg for intraocular tumors).
Round 3
Reviewer 2 Report
Comments and Suggestions for Authors
The authors did not address all my concerns regarding the lack of proper conjunctiva control. However, because indeed there is a lack of information regarding adrenergic innervation in pterygium, I decided that the article may be published after including the limitations statement: low number of conjunctiva analysis for genetic testing (only 4 conjunctiva) and lack of proper control.
Author Response
Thanks a lot for your comments, all of them are valuables for us. According to the last observation we included the following text in the manuscript in the methods: "
3.7. Observation limitations statement low number of conjunctiva analysis for genetic testing (only 4 conjunctiva) and lack of proper control."